# Health of Family Members of Road Transport Workers: Evaluation of Perceived Stress and Family Effectiveness

**DOI:** 10.3390/ijerph21101315

**Published:** 2024-10-02

**Authors:** Fernanda Lise, Mona Shattell, Raquel Pötter Garcia, Wilson Teixeira de Ávila, Flávia Lise Garcia, Eda Schwartz

**Affiliations:** 1Nursing Faculty, Federal University of Pelotas, Pelotas 96077-170, RS, Brazil; wilson.teixeira@ufpel.edu.br (W.T.d.Á.); edaschwa@gmail.com (E.S.); 2College of Health, Oregon State University, Corvallis, OR 97331, USA; 3College of Nursing, University of Central Florida, Orlando, FL 32826, USA; mona.shattell@ucf.edu; 4Anthropology Faculty, Federal University of Pelotas, Pelotas 96010-610, RS, Brazil; raquelgarcia@unipampa.edu.br; 5Nursing Faculty, Federal University of Pampa, Uruguaiana 96413-170, RS, Brazil; flgarcia@ufpel.edu.br; 6Nursing Faculty, Federal University of Rio Grande, Rio Grande 90040-060, RS, Brazil

**Keywords:** family health, health promotion, nursing, public health, systemic organization, stress, road transport workers, truckers

## Abstract

The health of road transport workers is affected by working conditions and life. However, there is a lack of studies on the level of stress and health of the families of these workers. This study aimed to evaluate the perceived stress level and family effectiveness of family members of road transport workers. A quantitative study was carried out with the family members of road transport workers in the southern region of Brazil. For data collection, a sociodemographic form, the Perceived Stress Scale (PSS), and the Evaluation of Family Effectiveness Strategies were used. The data were analyzed by simple frequency, Spearman correlation coefficient (ρ) (*p* < 0.05), and descriptive analysis from the perspective of Systemic Organization. The sample was composed of 49 family members of road transport workers. Perceived stress was higher in family members who had more than nine years of education (*p* = 0.0403). Family members who scored higher in Family Effectiveness scored high on the targets of Control (*p* = 0.0353) (Control aims to reduce anxiety and prevent and eliminate events that threaten family stability) and Growth (*p* = 0.0360) (represented by attitudes that promote new roles in response to critical situations experienced by families, which require re-adaptation processes and adjustments). The Control target was significant (*p* = 0.0353) in families that had more than three people. The Coherence dimension (concerning self-esteem, body image, personal identity, self-confidence, and sexual identity) presented positive significance (*p* = 0.0244) in families with health problems and whose income was less than USD 792.00 per month (*p* = 0.0072). The Individuation dimension (including functions and responsibilities, where talents are reinforced, as well as initiatives that allow for the incorporation of knowledge to assume behaviors against personal/family and environmental pressures), was significant (*p* = 0.0138) in families with incomes over USD 792.00. The Maintenance System (strategies for decision-making, problem negotiation, ritual and traditional roles, communication patterns, standards, financial management, and approaches to maintaining family harmony) presented positive significance (*p* = 0.0151) in families where drivers worked as intercity drivers, as did the Stability target (*p* = 0.0196) (concerning the continuity of routines, structure, organization, traditions, and values assumed by the family and transmitted from generation to generation, which promote unity and the development of values, attitudes, and beliefs). In conclusion, social factors, such as education, income, diseases, type of worker activity in road transport, and number of people in the family, influenced perceived stress and family effectiveness, which demonstrates the need to increase the promotion of health care for the families of road transport workers.

## 1. Introduction

The health of road transport workers may be affected by behavioral, environmental, or occupational risk factors, considering that the work of urban bus drivers or long-haul truck drivers (LHTDs) involves repetitive, monotonous, stressful activities, work overload, long journeys, long-term work, irregular hours, pressures regarding schedules, insecurity due to violence, and spending long periods away from home [1,2,3,4,5,6]. 

Consequently, these common factors among road transport workers increase the risk of illness due to chronic diseases such as hypertension, diabetes mellitus, and obesity [6,7,8] and change the work–life–family balance [9] in ways that can influence the quality of family relationships and impact well-being [10].

### 1.1. Family Systems

Family system studies date back to the General Theory of Systems, which was developed by Austrian Biologist Ludwig von Bertalanffy and widely adopted by Gregory Bateson, an anthropologist, epistemologist, cyberneticist, and founder of Family Therapy and Systemic Thinking [11]. Bateson, in 2024, celebrates 120 years of his birth, and although his studies on family communication date back to the 1950s, they have been enthusiastically revisited by researchers in anthropology, nursing, communication, and psychiatry [12]. This is probably due to interest in the opportunities for interpreting interactions that emerge from conversations with families, provided by the systemic approach.

From the systemic perspective, the family is a complex system of mutual interaction, consisting of members (subsystems), inserted in other larger systems [13]. These individuals who have decided to live and interact emotionally with the main objective of supporting each other may not have consanguinity but are emotionally linked through resources and common or complementary talents that allow them to fulfill certain roles that contribute to the functioning of the family as a total unit and to the constant exchange with the environment [14]. Thus, family functioning refers to the set of interpersonal relationships that occur in each family, which gives them their own identity [14].

Based on cultural factors, arrangements and family roles are socially defined, and families are organized to meet the health needs of their members. From the systemic perspective, family health is seen as a dynamic process, whose effectiveness can be evaluated with the use of theoretical instruments and models [14], as well as its structure through different family arrangements [15]. Its development can be classified into cycles, with each stage possessing unique characteristics proportional to the professional understanding of the stage(s), allowing families to be guided regarding the characteristic challenges of a certain stage of the life cycle [16]. This approach enables health professionals to understand the organizational structure of families and their operating systems [17], based on the commitment and competencies developed to support families during the care of individuals and their families at different levels of health care.

### 1.2. The Framework of Systemic Organization

The Framework of Systemic Organization was developed by the North American nurse Marie-Luise Friedemann. The framework aims to achieve four targets to find congruence (harmony). However, congruence is not fully achieved, being a constant search for the maintenance of family health. Thus, family effectiveness is the result of a set of situations and conditions, which include the individual and the environment, since the assessment of a family’s health level concerns a quality that indicates how a certain function is developed. Therefore, functionality is considered a health component, considering the environment as a facilitator or a barrier to activity performance [14].

To measure how the family is organized to function effectively as a system and to respond to the demands of each member and the environment, Dr. Friedemann developed an instrument, The Assessment of Strategies in Family Effectiveness, ASF-E, supported by the Framework of Systemic Organization [18], which has been used to assess family effectiveness in different contexts, including Germany, Brazil, Colombia, Chile, Finland, Mexico, Portugal, and Switzerland [19].

To respond to system requirements, the family function is maintained by the values and beliefs developed through family processes, including System Maintenance, System Change, Individuation, and Coherence, and dimensional targets, including Stability, Growth, Control, and Spirituality. The dimensional targets are subscales in the ASF-E [14]. Each family process is comprised of two-dimensional targets: System Maintenance comprises stability and Control, System Change comprises Growth and Control, Individuation comprises Growth and spirituality, and Coherence comprises Spirituality and Stability [14]. The System Maintenance process is about decision-making strategies, problem negotiation, ritual and traditional setting, communication patterns, standards, values, financial management, and an approach to the future to maintain family harmony that provides the family with a sense of security and autonomy. The System Change process refers to family actions to incorporate new events, such as the family’s preparation to receive a new member, move to another city, or retire, as the family’s priorities change over time. The Individuation process is related to personal identity including roles and responsibilities, where talents, initiatives, and knowledge are reinforced in reaction to personal/family pressures and the environment and by attitudes assumed by family members to achieve particular interests and make sense of life. The Coherence process considers harmonious relationships between family members by providing a sense of unity and family belonging through the internalization of respect, love, and concern for others, sharing values and beliefs that allow for the creation of emotional connections necessary for the survival of the system. Coherence also includes attitudes and strategies to maintain affective ties in shared activities and must correspond to the definition of values and principles established by the family [14].

### 1.3. Characteristics of Road Transport Workers’ Families

There is limited research addressing family health from the perspective of the family members of road transport workers, exploring the workers’ perspectives on their lifestyles, risk of diseases [20], work–life–family balance [9,21,22], psychological stressors, such as loneliness and loss of family conviviality [23], and the relationship with the family and coping strategies to overcome family isolation [24]. Characteristics of the families of road transport workers were described in studies carried out with LHTDs. The majority of LHTDs were married [25,26,27], belonged to nuclear families (88.4%) [9], had families composed of three or more members (66%) [27], and had an average yearly income of USD 80,000 (54%). However, regarding yearly income, about 95% of it was used for truck maintenance and work-related expenses [28]. These studies also revealed that road transport workers worked more than 13 hours per day (37.5%) [5], and for long-haul drivers in particular, the period away from home ranged from more than 14 days (33%) [2] to more than 21 days (84%) [7]. Because of this, these workers lost family conviviality and had feelings of loneliness, which was responsible for increased stress and anxiety [23].

### 1.4. Mental Health and Well-Being of Road Transport Workers and Families

In our society, numerous biopsychosocial and economic factors can affect people’s well-being and mental health, contributing to the increase in the burden of chronic non-communicable diseases [29]. For road transport workers, such as LHTDs, military, pilots, sailors, and flight attendants, among others, whose work activities and rhythms are intense and require them to be away from home for long periods, stressors may be related to a lack of work–life balance [7,9].

The imbalance between work and personal activities hinders the development of other social roles, such as participation in family activities, leisure activities, and other promoters of well-being and health. Consequently, the physical and psychological health of LHTDs is harmed, and this leads to a loss in the quality of sleep, contributing to the development and aggravation of cardiovascular diseases and mental health problems [5,7,26]. In addition, social relations with family and friends are also affected [5], possibly because of a lack of time for other activities, an overload of family members with household tasks, and the education of children.

Regarding road transport workers’ everyday work schedules and their implications for their health and the health of their families, it is relevant to evaluate the stress and health of their families, considering that stress can negatively impact health and contribute to the development of chronic diseases [29]. According to the theoretical model of Systemic Organization [14], stressful situations can destabilize the harmony of family systems, and the balance of relationships is fundamental to reducing stress.

This study presents factors that are associated with the perceived stress and family effectiveness of Brazilian family members of road transport workers. Perceived stress and family effectiveness may be impacted by the physical and mental stressors that interfere with the health of road transport workers [2,7,8,9,21,22,23]. In this study, the hypotheses are (1) the level of stress and family effectiveness of the family members of road transport workers will be influenced by social determinants of health such as years of education, income, diseases, and the number of people in the family and (2) the level of stress and family effectiveness of the family members of road transport workers will not be influenced by social determinants of health such as years of education, income, diseases, and number of people in the family. This study aimed to evaluate the perceived stress level and the family effectiveness of family members of road transport workers.

## 2. Materials and Methods

### 2.1. Study Design, Participants, and Setting

This cross-sectional, quantitative, and descriptive study was conducted from a convenience sample of 49 family members of road transport workers from the southern region of Brazil. Data collection took place from March to July 2023.

Family members self-identified as the families of LHTDs (and these family members could have been related by blood or could not have been related by blood). The inclusion criteria were the following: a family member of a road transport worker who was a driver of an urban bus or an intercity and/or LHTD, 18 years of age or older, and could read Portuguese. Family members of retired road transport workers, family members of road transport workers who are no longer living, family members 17 years or younger, and non-family members of LHTDs were excluded.

The Brazilian Health System (called the Unified Health System—SUS) offers free health care, at different levels, where nurses perform consultations and home visits to individuals and families. The health service where the data collection occurred is not linked to SUS; it provides health care to road transport workers and their families for a small fee. The health service is located in the south of Rio Grande do Sul, Brazil, and providers include a general physician, urologist, gynecologist, pediatrician, and dentist. This place was selected for the ease of access to the study population.

Family members were approached by the researcher (first author), accompanied by three nursing graduate students and a psychology undergraduate student, all of whom were trained in data collection. Participants were approached in the waiting room. Only one family member per family could participate, and these participants were the family members who had health service appointments. The objectives of this study were presented, and the type of participation was explained. The Free and Informed Consent Form was read, and those who agree to participate signed two copies, one for the researcher and the other for the participant. Data collection instruments were given to the participants to complete on their own; however, some needed support reading them. Although precise numbers were not recorded, we estimate that approximately 10% of potential candidates declined to participate because of the time required to complete the questionnaire.

### 2.2. Measures

Data collection tools included a sociodemographic form containing 13 items, the Perceived Stress Scale (SSP) with 14 items, and the Assessment of Strategies in Family Effectiveness (ASF-E/Brazil) instrument with 20 items, for an overall total of 47 items. Each participant completed the questionnaires on paper and took an average of 20 min to complete the data set. The data were initially entered in a Google form. Then, the data were exported to an Excel file that was later imported into the Statistical Analysis Software Program (SAS) for analysis.

The Perceived Stress Scale (PSS) was developed and validated for use in Brazil [30]; its reliability was obtained by internal consistency, in which Cronbach’s α = 0.82. The PSS measures the intensity at which people perceive situations as stressful, so it measures perceived stress. It is one of the most cited scales in the literature for estimating stress and can be used in different age groups because it does not have questions with contextual specificity. For example, one of the items is This last month, how often have you been upset because of something that happened unexpectedly? Answers are marked by a Likert scale with five points (0—never, 1—almost never, 2—sometimes, 3—often, and 4—very frequent), and the total results may vary between 0 and 56 (0 indicating no perceived stress; 56 indicating high perceived stress). Half of the items are positive situations and half are negative situations. To obtain the final score, the responses to items 4, 5, 6, 7, 9, 10, and 13 are reversed as follows 0 = 4, 1 = 3, 2 = 2, 3 = 1, 4 = 0. The negative responses are added directly. All the responses are added together for a final score. The higher the sum, the higher the perceived level of stress [31]. The authors of the PSS do not recommend dividing the scores into levels of perceived stress, such as low, medium, and high, because of loss of precision in statistical analysis [30].

The Assessment of Strategies in Family Effectiveness (ASF-E) was developed by the American nurse, Marie-Luise Friedemann [18]. The ASF-E instrument was developed for use with multicultural families at various educational levels; its reliability was obtained by internal consistency, in which Cronbach’s α ranged from 0.60 to 0.80. It measures family health (effectiveness) and has 20 items. It has 4 subscales, which refer to the targets of the framework, including Stability, Growth, Control, and Spirituality. The instrument is based on the theoretical model of Systemic Organization [14]. The ASF-E/Brazil was adapted to Brazilian Portuguese and validated to be used with Brazilian families [32]. It is a self-administered instrument, and it measures the level of health or family functionality. Each item has three choices worth one, two, or three points, where higher points indicate greater family effectiveness as follows: three points indicate high family effectiveness; two points indicates average family effectiveness; and one point indicates low family effectiveness. For example, item 1 of the instrument has the following respective options: “There is anger or sadness in our family” (scored 1); “People in our family do not openly express their feelings” (scored 2); and “Our family is happy, in general” (scored 3). The instrument has a minimum value of 20 points and a maximum total value of 60 points. The overall classification is as follows: high level of family effectiveness with a total score between 48 and 60; intermediate level of family effectiveness with a total score between 34 and 47; and low effectiveness of family functioning with a total score between 20 and 33 points [14,18,32]. The 20 items of the ASF-E/Brazil are classified into four dimensions as follows: Coherence (item = 1, 10, 15, 18), Individuation (items 3, 5, 7, 12), System Maintenance (items 4, 8, 13, 17, 19), and System Change (items 2, 6, 9, 11, 14, 16, 20). These dimensions comprise the following four targets of the family system: Stability target (composed of the System Maintenance and Coherence dimensions) (items 1, 2, 6, 9, 10, 11, 14, 15, 16, 18, 20); Control (System Maintenance and System Change) (items 2, 4, 6, 8, 9, 11, 13, 14, 16, 17, 19, 20); Growth (Individuation and Change in the System) (items 3, 4, 5, 7, 8, 12, 13, 17, 19), and Spirituality (Coherence and Individuation) (items 1, 3, 5, 7, 10, 12, 15, 18).

### 2.3. Data Analysis

The data were analyzed in SAS, version 9.4, from a database built from the Excel application. The data were described in tables of absolute numbers and frequency distributions. To measure the difference between the scores attributed to latent variables and sociodemographic variables, the sum score was used for each variable and the Wilcoxon–Mann–Whitney or Kruskal–Wallis test (ANOVA) was applied, followed by the Dunn post hoc multiple comparisons test for the cases in which the observed variable had more than two categories. A confidence level of 95% (α = 0.05) was set.

### 2.4. Ethics Considerations

In this study, ethical principles of research involving human beings were respected, according to the Resolution of the National Health Council No. 466 of 12 December 2012. The Free and Informed Consent Form was delivered and read to the participants on the day of data collection, and two signed copies were obtained for the participant and the researcher, ensuring the freedom of spontaneous participation and the right to withdraw at any time. This study was approved by the Ethics Committee of the Federal University of Pelotas, Nursing Faculty (protocol code 5,892,602 and Certificate of Presentation for Ethics Assessment number 66722622.9.0000.5316) on 14 February 2023.

## 3. Results

### 3.1. Characteristics of the Sample

The questionnaires were completed by 49 family members of road transport workers. Sociodemographic characteristics of family members of road transport workers showed that the majority were female (97.96%), of which many (89.80%) were wives and most were married (87.76%). The mean age group was 40 years (61.22%); most had nine years or more of education (73.47%) and self-declared as white (73.47%). Religious affiliations included Catholic (32.65%), evangelical (24.49%), without religion (16%), spirit (16%), traditional African (4%), and other (6%). Health problems were reported by 38% of the family members of road transport workers (22% reported having a diagnosis of hypertension, 8% diabetes mellitus, 6% depression, 4% hyperthyroidism, 2% anxiety, asthma, and spine problems); some participants cited more than one health problem. Regarding the work modality of the family member’s road transport worker, almost half (48.98%) of the family members reported that their family members were working as drivers in the urban sector, followed by long-haul truck drivers (32.65%); only 18.37% were intercity drivers. Slightly more of the family member’s road transport workers in the sample transported people (51.02) as opposed to loads (48.98%). See Table 1 for a full description of the sociodemographic characteristics of our sample.

The average family income was BRL (Brazilian Real) 4276.73 or USD 855.34. In each family residence, there were 3.24 residents on average, resulting in a per capita income of BRL 1431.93 or USD 286.38; two-thirds (64.58%) received between 1 and 3 times the minimum wage. Three-quarters (73.47%) of the family residences included three to four people.

### 3.2. Perceived Stress

Perceived stress was measured using the 14-item Perceived Stress Scale. Total scores on the PSS in our sample ranged from 7 to 38, with a mean of almost 24 (23.9), which indicates that our sample had some level of perceived stress. See Table 2 for the frequency distributions from the PSS.

### 3.3. Family Effectiveness

Family effectiveness was measured using the Instrument Assessment of Strategies in Families—Effectiveness (ASF-E/Brazil). Most participants scored level 3, corresponding to high family effectiveness. For level 3, the highest percentage was about special dates (95.9%; item 19), followed by satisfaction with the place where they live (89.8%; item 4). For the level 2 category, indicating average family effectiveness, the highest percentages were recorded for family pride (83.6%; item 5), and participation in community activities (61.2%; item 7). For level 1, corresponding to the lowest level of family effectiveness, the highest percentage was related to family consultation to make personal decisions (53.0%; item 17). See Table 3 for the distribution of responses to the ASF-E/Brazil on family effectiveness.

No significant differences were found in the perceived stress, dimensions, and targets between the family group up to 39 years old (n = 19) and the age group. For marital status, there were no significant differences between those who were married (n = 43) and those who were not (n = 6). The number of residents in the family, up to two people (n = 10) and more than two people (n = 39), did not change the perception of the participants in relation to perceived stress and sociodemographic characteristics. For the level of education, a significant difference was observed (*p* = 0.0403) for perceived stress. Participants (n = 36) with 9 years or more of education had a higher level of perceived stress. See Figure 1 for the mean perceived stress scores by sociodemographic characteristic.

Families with family income that was between 1 and 3 times the minimum wage showed a higher average level (*p* = 0.0072) in the Coherence dimension. Participants with an income greater than 3 times the minimum wage had a higher average score for the Individuation dimension (*p* = 0.0138). Participants with some health problems (n = 15) had a higher level of perception (*p* = 0.0244) in relation to the Coherence dimension. The Coherence dimension also reflects concern for others. The driver work type presented a significant difference (*p* = 0.0151) for the System Maintenance dimension. This dimension brings together strategies to maintain family harmony, providing a sense of security and autonomy. The family members of the intercity drivers presented a higher average level of perception than LHTDs in relation to the System Maintenance dimension (Figure 2).

The education variable showed significant differences between the Control (*p* = 0.0325) and the Growth targets (*p* = 0.0360). Participants who had 9 years or more of education had a higher average level of perception for these two dimensions. The Control target was also influenced by the number of residents. The Control target was significant (*p* = 0.0353) in the residences, whose number of residents was equal to or greater than three people. The type of work activity variable presented a significant difference (*p* = 0.0196) for the Stability target. Family members of intercity drivers presented a higher average score than family members of drivers in the urban sector and long-haul truck drivers. This target results from the Coherence and System Maintenance dimensions (Figure 2).

## 4. Discussion

This study presented how select factors—years of education, family income, the number of members of the family group, the occurrence of health problems, and the type of activity performed by the worker, as a driver—influence the perceived stress and family effectiveness of family members of road transport workers using the Framework of Systemic Organization. The results may be related to family contexts, social and environmental conditions, the complexity of family relationships, the people who make up the family, in the post-COVID-19 pandemic period, and the work of road transport workers.

There was a predominance of female participants. Because drivers are mostly male [2,5,26,28], the participants were mainly the female wives of male road transport workers (89%) who attended the health service where data collection was conducted. Similarly, our sample consisted of mostly women 40 years or older, corresponding to the average age of road transport workers, which is 45 years or older [5,8,26].

The years of education showed a significant difference in the perceived stress of the road transport workers’ family members, demonstrating that the higher the education level, the greater the perceived stress. Considering that stress is a natural human biological response that prompts us to address challenges and threats in our lives, it is possible that overloading people’s lifestyle activities with more years of education influenced their perception of stress. This result demonstrates the need to support individuals and families in stressful and deteriorating situations of the family system. On the other hand, in the evaluation of family effectiveness, the Growth and Control targets showed a significant difference in the years of education, with greater family effectiveness for participants with more than nine years of education. The significant difference in the years of education in the Growth target is consistent, considering that this target results from the dimensions of Individuation and System Change. It is represented by attitudes that promote new roles in response to critical situations experienced by families, which require adaptive processes and adjustments to promote family health. The years of education variable demonstrates an influence on the processes of readjusting beliefs and attitudes that allow humans to explore new behaviors as a healthy response to a crisis, which leads to Growth [14]. In this sense, people with less education may need greater attention from family nurses to evaluate and support the family in achieving the balance of the system to promote the health of a worker’s family.

Education also had significant results for the Control target, which was higher in people with more than nine years of education. The Control target aims to reduce anxiety and prevent and eliminate events that threaten family stability, thus breaking congruence or harmony [14]. In this study, items 19 and 4 presented the best results at level 3 for this target. Item 19, related to the way the family organizes itself for special dates, with 95.9%, demonstrated that families value their rituals. Item 4, on satisfaction with the place where they live, with 89.8% at level 3, demonstrated appreciation for people and the place where they live. The worst result of the Control target was for item 17, with 53% at level 1. Item 17 corresponds to family consultation in personal decisions, demonstrating that the family members of road transport workers believe that they need to communicate and ask for family approval to make decisions. Previous studies have shown that the Control target protects against external and internal threats to the family system [33] and promotes a sense of inner security by reducing the anxiety produced by insecurity and vulnerability because it has strategies that allow them to make the necessary adjustments for the maintenance of the system [34]. For family members of road transport workers, communication can be considered a strategy to reduce anxiety. A study with truck drivers identified that communication with family and friends is frequent and often occurs during hours off [25].

The number of people in each family showed a significant difference in the evaluation of family effectiveness—the Control target was more effective in families composed of three or four members. This result may be related to the presence of children in the family, as this goal aims to reduce anxiety and prevent and eliminate events that threaten family stability or break with harmony, demonstrating the concern to prevent and eliminate threatening events [14]. In this context, these events may be related to the routine of LHTDs, as described in other studies with irregular, extensive journeys, long periods away from home, and insecurity due to urban violence [2,5,25]. Yosefl (2021) evaluated the family composition of road transport workers and found that most families were composed of three or more members (66%) [27]. Most were nuclear families (88.4%), and the maintenance of the family was considered the largest motivation of drivers (35%) to continue working [9].

The existence of one or more health problems was reported by 38% of the participating family members, including chronic diseases such as hypertension, diabetes mellitus, and depression, which, according to the World Health Organization, affect two billion people and cause three-quarters of the deaths worldwide [29]. A previous study reported higher levels of depression, anxiety, and stress, and whose family members’ work activities were in remote places [35]. Presenting some health problem was significant for the Coherence dimension, which concerns the stable components of a person, the body and its organs, self-esteem, body image, personal identity, self-confidence, and sexual identity built since childhood and with reference to parents. This dimension is part of the Spirituality and Stability targets and therefore provides a sense of unity and family belonging through the internalization of respect, love, concern for the other, and sharing values and beliefs that allow for creating the emotional connections necessary for the survival of the system [14]. However, the reach of Coherence is different among people and can reduce tension with physical exercise, body consciousness practices, listening to music, appreciating nature, participating in cultural and artistic activities, meditating, and others. This result may be related to the strategies used by families to care for family members with chronic disease, demonstrating that they have resources that allow for strengthening the emotional bonds that correspond to the values and principles that govern the family system.

The average family income was BRL 4276.73 or USD 831.00 per month. Of these participants, two-thirds (64.58%) received a salary that was 1 to 3 times the minimum wage, corresponding to up to BRL 3960.00 or USD 760.00 per month. Previous studies with drivers showed that income was less than BRL 1133.00 or USD 220.00 per month (YOSEF et al., 2021) and the absence of appropriate increases in earnings caused additional stress in drivers because they were not able to provide adequate financial resources to their families [20]. In addition, financial problems prevented family members from obtaining health care [28]. In the Coherence dimension, the significance was higher for families receiving up to three times the minimum wage. In this dimension, item 10, related to family work, had the worst result, with 49% at the low and intermediate levels. This result may be related to the characteristics of the sample, composed of women, wives, and daughters of road transport workers, revealing the need for support and participation of other members in the household for family tasks and work. The item with the best result was number 1, related to family happiness, with 75.5% at the high level, which, according to Systemic Organization, demonstrates the connection between the members and that allows for the survival of the family system [14].

On the other hand, in the Individuation dimension, the significance was higher in families whose income was higher than three times the minimum. This dimension is a structure in which personal identity is developed, which includes roles and responsibilities, and where talents are strengthened, as well as activities that allow knowledge to be incorporated to assume behaviors against personal/family pressures and the environment. The Individual dimension is represented by attitudes assumed by family members to achieve interests and to give meaning to life [14]. For this dimension, the items of ASF-E/Brazil with the worst results were 5 and 7, which are related to family pride, item 5, with only 2% at a high level, and participation in community activities, item 7, with 22% at a high level. The best results were for items 3, related to help, and 12, related to solving problems, with 81.6 and 77.5% respectively, at a high level.

The lower family pride and low involvement in community activities may be related to cultural aspects. As described in previous research, road transport workers perceive a lack of recognition of truck driver activity and poor image of this professional category because of a low salary, lack of career progression, and most likely, time away from family and home [36]. As intellectual and physical activities expand personal horizons, recognizing prejudice and increasing social support networks can strengthen pride and participation in community activities. Attitudes such as the availability to help in the community, identified in item 3, and the ability to solve problems, identified in item 12, favor the survival of the family system through the adjustments of values.

Among the type of driver work, the families of the intercity drivers, who carry people or cargo, presented a significant difference in the evaluation of family effectiveness, in the System Maintenance dimension, and in the Stability target. The work of intercity drivers involves the transportation of cargo or people, with the characteristic, also common to urban drivers, of returning home daily at the end of each work day. However, LHTDs may face greater stress in interpersonal relationships due to long periods of home absence and long and irregular hours. Prior studies evaluated the impact of long periods of home absence, due to worker activity in remote places, on family health and revealed high levels of depression, anxiety, and stress in workers’ families, who presented great concern [35]. Another study found that the distancing of fathers caused greater impact on the mental health of the children of a military father [37].

The System Maintenance dimension uses strategies for decision-making, negotiating problems, establishing rituals and traditions, and defining roles, communication patterns, norms, values, financial management, and future approaches to maintaining family harmony, providing the feeling and security and autonomy. This dimension obtained better results for items 9 and 20. Item 9, related to how the family faces problems, with 87% at level 3, demonstrated better management in solving problems. Item 20, related to freedom, with 85% at level 3, showed that members feel free to be themselves. The Stability target also showed significant differences in the activities of the family members of intercity drivers. This target results from the dimensions of Coherence and System Maintenance and concerns the continuity of routines, structure, organization, traditions, and values assumed by the family and transmitted from generation to generation, which promote unity and development of values, attitudes, and beliefs [14]. This result is related to the lowest level of anxiety, possibly related to the opportunity to return home each day, which improves communication patterns and favors the fulfillment of roles experienced by the families of intercity drivers.

From the results obtained in this study, it is possible to affirm that the type of activity performed by the worker in road transport affects the health of the family, which is better for family members of workers in intercity transport, whose activity allows the worker to return home each day. The work routine of long-distance drivers and buses in the urban sector involves a stressful work process, irregular times, high workload, and fear of violence, which can affect personal relationships by compromising communication, provoking anxiety, and hindering the driver’s participation in family events and activities.

A previous study showed that worker health was affected by the imbalance between work and family and led to a higher level of sleep-related problems [22]. In addition, the need to work at times that do not coincide with those of family and friends, such as during the overnight hours, can be considered a stress trigger [21] and can compromise the workers’ support systems, as a result of long family separations, work demands, lack of energy when they are at home, and often lead to a breakdown of relationships or divorce [24]. For drivers, because of the nature of the work, which requires long hours and last-minute demands, separation from their families resulted in guilt for needing to stay away from home for long periods, compromised parenting, and the loss of monitoring the growth of their children [20,21], and it also prevented participation in family events [20,24].

### 4.1. Implications for Practice

The families of road transport workers need to adapt to the routines of irregular schedules, absences for long periods, and concerns about the health and safety of their relatives while they are at work. These factors need to be considered by nurses and other healthcare workers to evaluate the health of the families of workers in urban and or long-haul transport and intervene to meet their needs. The findings from this study can guide the health care of this high-risk population for chronic diseases, which is due to stress and other factors that involve working and living conditions. Considering that family harmony is disrupted by situations of stress or imbalance in people’s relationships, evaluation approaches using instruments, such as the Perceived Stress Scale and the Evaluation Instrument of Family Effectiveness Strategies (ASF-E/Brazil), supported by the systemic approach can guide interventions to promote stress reduction and improve the health of the family by contributing to increased family congruence.

### 4.2. Limitations

The limitations of this study are the small sample size, the descriptive analysis, and the cross-sectional design. Another limitation of this study is the absence of information that would have allowed the authors to describe the organization, types, and ages of persons in the family, the stage of the family cycle, the origin of the family income sources, and the working time in road transport of the family member of the participants. However, this did not reduce the quality of the findings about perceived stress and family effectiveness. More research is needed to explore this and other aspects related to the health of workers and their families in the prevention of chronic diseases.

## 5. Conclusions

This study evaluated the perceived stress level and the family effectiveness of the family members of road transport workers. It showed that perceived stress was higher in family members with more than nine years of education, as opposed to the ASF-E/Brazil instrument, in which Control and Growth targets were positively influenced by the education variable. The Control target was higher in families with more than three people in the family. Income positively influenced the dimensions of Coherence and Individuation, and the Coherence dimension was also influenced by the health problems variable. The driver work type positively influenced the System Maintenance dimension and the Stability target for workers in intercity transport.

These results allow us to infer that higher education, despite increasing perceived stress, allows families to create strategies to maintain family harmony. The activity of the family member as an intercity driver allows drivers to return home each day, improving the health of the family, possibly because of the greater ability to seek solutions to adversity and the availability of the driver to engage in family activities. This demonstrates the ability of family members to adjust and create strategies to deal with adverse situations, such as health problems and stressors, and to adjust to the needs of members to maintain family harmony. In addition, it indicates the need for greater support and professional attention to the health of workers’ families in urban areas and those of long-haul truck drivers, especially in relation to perceived stress, to create strategies that will reduce stressors. However, it is not possible to generalize these results, and more research is needed to better understand the influence of sociocultural factors of work and life on the health of families.

## Figures and Tables

**Figure 1 ijerph-21-01315-f001:**
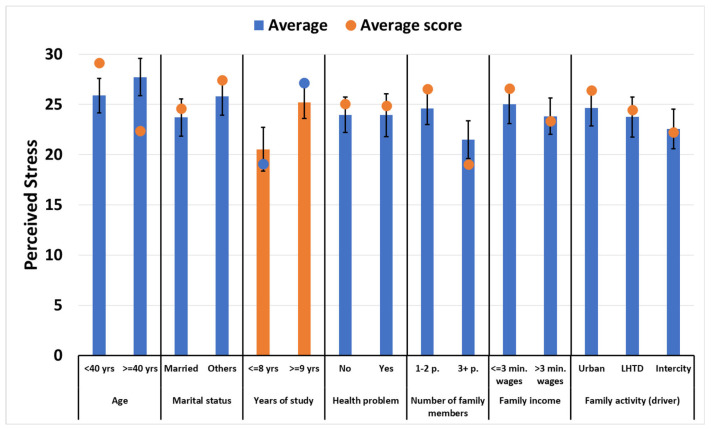
Mean (standard deviation) of the perceived stress level of the family members of road transport workers. Pelotas, RS, Brazil, 2023. Significant at the 95% confidence level; SSP = Perceived Stress Scale.

**Figure 2 ijerph-21-01315-f002:**
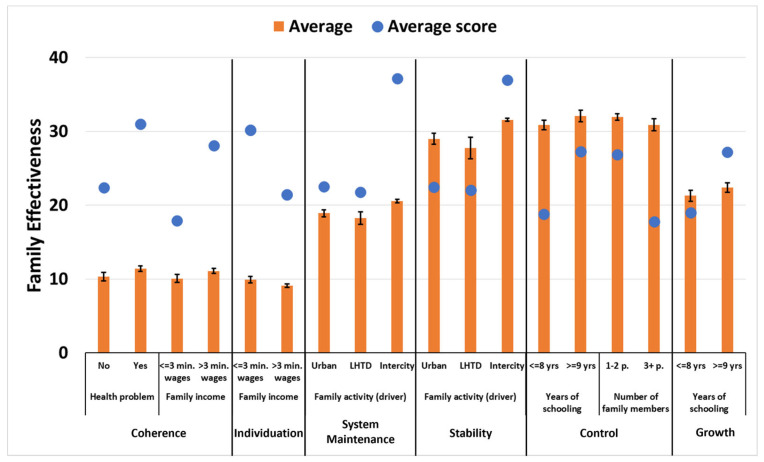
Mean (standard deviation) and mean score of targets and dimensions of the Family Effectiveness Assessment Instrument (ASF-E/Brazil) of the family members of road transport workers. Pelotas, 2023. (Only results with significance). Significant at the 95% confidence level; ASF-E/Brazil = Assessment of Strategies in Families—Effectiveness.

**Table 1 ijerph-21-01315-t001:** Sociodemographic characteristics of the family members of road transport workers, Pelotas, RS, Brazil, 2023.

Features	N	%
Sex		
Female	48	97.96
Male	1	2.04
Age group		
Up to 39 years	19	38.78
40 years +	30	61.22
Degree of kinship		
Wife	44	89.80
Others *	5	10.20
Civil status		
Married	43	87.76
Others **	6	12.24
Years of education		
Up to 8 years	13	26.53
9 +	36	73.47
Ethnicity		
White	36	73.47
Other ***	13	26.53
Religion		
Catholic	16	32.65
Evangelical	12	24.49
Other ****	21	42.86
Family income *****		
From 1 to 3 m.w.	31	64.58
More than 4 m.w.	17	35.42
Health problems		
Yes	14	38.60
No	35	71.40
Number of residents		
Up to two	10	20.41
From three to four	36	73.47
More than four	3	6.12
Type of worker activity ******		
Intercity driver	9	18.37
Long-haul truck driver	16	32.65
Urban sector	24	48.98
Type of transport		
Loads	24	48.98
People	25	51.02

* Son/daughter, father, mother, brother/sister, ** S for union, single, widowed, divorced/separated, *** Black, brown. **** Spiritist, without religion, traditional African religion, ***** 1 Brazilian minimum wage in 2023 = BRL 1320.00 or USD 258.00 per month. ****** of the worker in road transport.

**Table 2 ijerph-21-01315-t002:** Frequency distribution of data from the Perceived Stress Scale (PSS) (n = 49) of the family members of road transport workers, Pelotas, RS, Brazil, 2023.

Item/Number	Mean	Standard Deviation	CV	Median	Minimum	Maximum
Item 1	1.80	1.02	56.81	2	0	4
Item 2	1.35	1.11	82.38	1	0	4
Item 3	2.22	1.01	45.20	2	0	4
Item 4	2.22	1.01	45.20	2	0	4
Item 5	0.94	0.90	95.79	1	0	4
Item 6	1.00	1.00	100.00	1	0	4
Item 7	1.67	1.14	68.34	2	0	4
Item 8	2.04	1.04	50.96	2	0	4
Item 9	1.24	1.05	84.46	1	0	4
Item 10	1.35	0.78	57.81	1	0	3
Item 11	1.86	1.14	61.20	2	0	4
Item 12	3.35	0.88	26.27	4	0	4
Item 13	1.59	1.17	73.57	2	0	4
Item 14	1.33	1.13	84.83	1	0	4
Total	23.96	6.45	26.91	24	7	38

**Table 3 ijerph-21-01315-t003:** General classification of items according to the levels of the dimensions of the ASF/Brazil instrument (n = 49) of the family members of workers in road transport, Pelotas, RS, Brazil, 2023.

Dimension/Item	Distribution of Responses ASF-E/Brazil			Total
	Level 1	Level 2	Level 3	
	n (%)	n (%)	n (%)	
Consistency (C)				
Item 1	3 (6.12)	9 (18.37)	37 (75.51)	49 (100)
Item 10	4 (8.16)	20 (40.82)	25 (51.02)	49 (100)
Item 15	6 (12.24)	16 (32.65)	27 (55.10)	49 (100)
Item 18	1 (2.04)	22 (44.90)	26 (53.06)	49 (100)
Individuation (I)				
Item 3	3 (6.12)	6 (12.24)	40 (81.63)	49 (100)
Item 5	7 (14.29)	41 (83.67)	1 (2.04)	49 (100)
Item 7	8 (16.33)	30 (61.22)	11 (22.45)	49 (100)
Item12	4 (8.16)	7 (14.29)	38 (77.55)	49 (100)
System Change (SC)				
Item 4	3 (6.12)	2 (4.08)	44 (89.80)	49 (100)
Item 8	3 (6.12)	8 (16.33)	38 (77.55)	49 (100)
Item 13	3 (6.12)	16 (32.65)	30 (61.22)	49 (100)
Item 17	26 (53.06)	12 (24.49)	11 (22.45)	49 (100)
Item 19	0 (0.00)	2 (4.08)	47 (95.92)	49 (100)
System Maintenance (SM)				
Item 2	3 (6.12)	10 (20.41)	36 (73.47)	49 (100)
Item 6	2 (4.08)	11 (22.45)	36 (73.47)	49 (100)
Item 9	3 (6.12)	3 (6.12)	43 (87.76)	49 (100)
Item 11	5 (10.20)	14 (28.57)	30 (61.22)	49 (100)
Item 14	1 (2.04)	8 (16.33)	40 (81.63)	49 (100)
Item 16	1 (2.04)	13 (26.53)	35 (71.43)	49 (100)
Item 20	2 (4.08)	5 (10.20)	42 (85.71)	49 (100)

## Data Availability

The data presented in this study are available upon reasonable request from the corresponding author.

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
