# Peer review of "Health of Family Members of Road Transport Workers: Evaluation of Perceived Stress and Family Effectiveness"

_ijerph, 2024, doi:10.3390/ijerph21101315_

Round 1

Reviewer 1 Report

Comments and Suggestions for Authors

Line 94

The manuscript basically made a clear elaboration on the concept of the four dimension related to family system. However, it would be much more better if the manuscript could exemplify what types of family actions could incorporate what kinds of new events when it comes to The System Change dimension, just like what the manuscript did in the narratives of The System Maintenance dimension.

From Line 45 to Line 136

Based on what I have learned in the chapter INTRODUCTION, the farming here was organized by the family relationships and its assessment from the perspectives of systemic way. Hence, the manuscript did some literature review on that topics. But, it turns to mental health issues so rapidly which can not manifest the relationship between mental health issues and family effectiveness clearly.

And actually you have mentioned two a bit discrete but insightful points, for one is the research gap in the perspectives of family members (line 107) and the other one is stressful situations interfaced with family system(line 126). Obviously if the manuscript could provide more details on that then it would strengthen the conversations with the prior research and let readers know the own contributions of this research. Like are there some conclusions from the prior research in the perspectives of workers, or does it make some contributions to the perspectives of family members?

So my advice is to arrange more content about the prior findings about mental health or just simply highlight those by modifying the structure of introduction, which might connect the two main parts of your research more tightly before any readers enter into the empirical part.

From Line 137 to Line 158

Since the manuscript have mentioned it is not a probabilistic sample, which means it might be a result happened after selection bias especially when the study was carried out in the same place and the same year. So, the manuscript supposed to give more convincing explanations on its representativeness or validity under this sampling strategy.

Also, there is another way to correct the potential bias and reduce the impact by potential confounding factors. The author might take more regional factors into consideration by introducing the local policy, local culture, local nursing conditions, etc. By doing this, the research might come with more insightful discussion and give answers to the general concern about the sampling strategy.

Author Response

Dear Reviewers,

We thank you for your contributions to the paper. They were important to improve the quality.

Reviewer 1

 The manuscript basically made a clear elaboration on the concept of the four dimension related to family system. However, it would be much more better if the manuscript could exemplify what types of family actions could incorporate what kinds of new events when it comes to The System Change dimension, just like what the manuscript did in the narratives of The System Maintenance dimension.

 Authors: Thank you for the feedback. We added text about the System Change dimension as suggested (see lines 166-167).

The System Change dimension refers to family actions to incorporate new events inside (as the preparation to receive a new member in the family or for moving to another city or retirement), as their priorities change over time.

Reviewer 1

From Line 45 to Line 136

Based on what I have learned in the chapter INTRODUCTION, the farming here was organized by the family relationships and its assessment from the perspectives of systemic way. Hence, the manuscript did some literature review on that topics. But, it turns to mental health issues so rapidly which can not manifest the relationship between mental health issues and family effectiveness clearly.

And actually you have mentioned two a bit discrete but insightful points, for one is the research gap in the perspectives of family members (line 107) and the other one is stressful situations interfaced with family system (line 126). Obviously if the manuscript could provide more details on that then it would strengthen the conversations with the prior research and let readers know the own contributions of this research. Like are there some conclusions from the prior research in the perspectives of workers, or does it make some contributions to the perspectives of family members?

So my advice is to arrange more content about the prior findings about mental health or just simply highlight those by modifying the structure of introduction, which might connect the two main parts of your research more tightly before any readers enter into the empirical part.

Authors: Thank you for the feedback. We provided mental health information, which we hope makes the connection clearer (see lines 194-247)

1.4 Mental health and well-being of workers and families

In our society, numerous biopsychosocial and economic factors can affect people's well-being and mental health, contributing to the increase in the burden of chronic non-communicable diseases [29]. Workers such as LHTDs, military, pilots, sailors, flight attendants among others, whose labor activity of intense rhythm, requires leaving home for long periods, stressors may be related to lack of work-life balance [7, 9].

The imbalance between labor and personal activities, hinders the development of other social roles, such as participation in family activities, leisure activities and other promoters of well-being and health. As a consequence, physical and psychological health is harmed and this leads to loss in the quality of sleep, contributes to the development and aggravation of cardiovascular diseases and mental health problems [5, 7, 26]. In addition, social relations with family and friends are also affected [5], possibly due to lack of time for other activities, difficulties of communication, overload of the family member with household tasks and education of children.

Reviewer 1

From Line 137 to Line 158

Since the manuscript have mentioned it is not a probabilistic sample, which means it might be a result happened after selection bias especially when the study was carried out in the same place and the same year. So, the manuscript supposed to give more convincing explanations on its representativeness or validity under this sampling strategy.

Also, there is another way to correct the potential bias and reduce the impact by potential confounding factors. The author might take more regional factors into consideration by introducing the local policy, local culture, local nursing conditions, etc. By doing this, the research might come with more insightful discussion and give answers to the general concern about the sampling strategy.

Authors: Thank you for the feedback. We provided more information on the location, which we hope addresses this suggestion (see lines 278-283).

The Brazilian Health System (called the Unified Health System - SUS) offers health care for free, at different levels of health care, where nurses perform consultations and home visits to individuals and families. The health service where the collections occurred is not linked to SUS, nor offers nursing consultation. However, it offers services (charging a small fee) to road transport workers and their dependents. This data collection site was selected for the access it provided to the study population.

Reviewer 2 Report

Comments and Suggestions for Authors

The authors present a cross-sectional study of 49 family members of road transport workers. The aim is to analyse the role of some selected socio-demographic factors on perceived stress and family effectiveness. The results provide an insight into the factors that may play a role in the family health of this target group.

I think the authors should make some revisions to improve the overall quality of the manuscript.

English proofreading is also needed.

Title and Abstract

The abstract would benefit from a brief introduction to the context of the study, i.e. what conditions exist for truck driver families and why it is important to study them, especially the variables included in the study. This should be inserted before the statement about the lack of studies on this topic.

Line 22: it is not clear what the higher values of the two subscales refer to.

Introduction

Lines 45 and 71 have the same numbering.

Line 36: when using an acronym for the first time, the full name must appear first and then the acronym.

Line 69: the role of healthcare professionals is not clear. How can they come into contact with this target group and support them?

Lines 87-90: this paragraph is not clear. How do the targets differ from the subscales?

Lines 112-113: do the authors mean that the characteristics of truck drivers found in the studies in the literature are those listed below?

Lines 128-129: what does it mean that the study presents “previously unpublished factors”? Also, “are associated with or play a role in” would be more appropriate than “interfere”.

Lines 130-131: this sentence seems incomplete.

Lines 132 and 135: statements are made with two different objectives. Perhaps they should be summarised in two different hypotheses.

Materials and Methods

Line 143: since the authors defined “family” as members who did not necessarily have to be related by blood, was this criterion applied when selecting the sample?

Line 147: why was only one person per family selected?

How many potential candidates refused to participate in the study?

What was the minimum number of samples considered necessary to perform the analyses?

Line 159: this section can be labelled “Measures" as the data collection actually starts in the paragraph above.

For each of the scales used in the study, a sample item should be provided as well as the Cronbach’s alpha calculated in the study sample.

Lines 190 and 196: these two sentences can be combined into one sentence.

Line 104: this paragraph should be called “Data analysis”.

Line 218: there is a reference to an ethics committee, but it is not stated which institution.

Results

Line 230: it’s not clear what the information on Covid-19 vaccination doses does. Was this used in the analysis?

Using the full item descriptions in the tables would be more useful than the numbers.

Discussion

Line 306: this first statement should be reformulated. The aim of the study was to investigate which factors – from a series of selected factors – are associated with the two main variables. Other factors that were not considered could also play an important role.

Line 316: the sentence is not clear. What is meant by “the study time”?

Lines 316-318: the two sentences are not well connected. If the higher level of education was found to be a predictor of stress, what possible reasons do the authors give for this result?

Lines 333-335: this sentence needs to be reformulated.

Lines 363-366: is there data from studies that refer to the same target group or that share similar characteristics?

Line 423: the sentence is not clear.

Limitations

Limitations include the small sample size and the purely descriptive analysis. Further studies should take into account additional factors that would allow deeper statistical analysis. Another limitation is the cross-sectional design. Longitudinal studies would also be appropriate to gain a deeper understanding of the phenomenon. Diary studies, for example, could also be suitable for smaller samples.

Comments on the Quality of English Language

English proofreading is needed.

Author Response

Dear Reviewers,

We thank you for your contributions to the paper. They were important to improve the quality.

Reviewer 2

The authors present a cross-sectional study of 49 family members of road transport workers. The aim is to analyse the role of some selected socio-demographic factors on perceived stress and family effectiveness. The results provide an insight into the factors that may play a role in the family health of this target group. I think the authors should make some revisions to improve the overall quality of the manuscript.

English proofreading is also needed.

 Authors: Thank you for the feedback. We provided a revised version that has been reviewed and edited by a native English-speaking coauthor.

Title and Abstract

Reviewer 2

The abstract would benefit from a brief introduction to the context of the study, i.e. what conditions exist for truck driver families and why it is important to study them, especially the variables included in the study. This should be inserted before the statement about the lack of studies on this topic.

Authors: Thank you for the feedback. We added a brief introduction (see lines 14-16)

The health of the worker in road transport is affected by working conditions and life, however, there is a lack of studies on the level of stress and the health of families of these workers.

Reviewer 2

Line 22: it is not clear what the higher values of the two subscales refer to.

 Authors: Thank you for the feedback. We added explanations of the two subscales, values, and targets (see lines 22-41).

Perceived stress was higher in family members who had more than nine years of education (p= 0,0403). Family members who scored higher in Family Effectiveness had scored high on the targets of Control (p= 0,0353), which aims aims to reduce anxiety, prevent and eliminate events that threaten family stability; and Growth  (p=0,0360), that is represented by attitudes that promote new roles in response to critical situations experienced by families, which require re-adaptation processes and adjustments. The Control target was significante (p= 0,0353) in families that had more than three people. The Coherence dimension, which concerns self-esteem, body image, personal identity, self-confidence and sexual identity, presented positive significance (p= 0,0244) in families with health problems and whose income was less than $792.00 US dollars per month (p= 0,0072). The Individuation dimension, which includes functions and responsibilities, where talents are reinforced, as well as initiatives that allow the incorporation of knowledge to assume behaviors against personal/family and environmental pressures, was significant (p= 0,0138) in families whose income was over $792.00. The Maintenance System, which includes strategies for decision making, problem negotiation, ritual and traditional role, communication patterns, standards, financial management and approach to the purpose of maintaining family Harmony, presented positive significance (p= 0,0151) in families whose drivers’ activity was as an intercity driver, as well as in the Stability target (p= 0,0196). The Stability target concerns the continuity of routines, structure, organization, traditions and values ​​assumed by the family and transmitted from generation to generation, which promote unity and the development of values, attitudes and beliefs.

Introduction

Reviewer 2

Lines 45 and 71 have the same numbering.

Authors: Thank you for catching this. We have changed the second section number from “1.1” to “1.2” (see line 126).

Reviewer 2

Line 36: when using an acronym for the first time, the full name must appear first and then the acronym.

Authors: Thank you for the feedback. We added the full name and put the acronym in parentheses immediately following (see lines 90-91).

Long-Haul Truck Drivers (LHTDs)

Reviewer 2

Line 69: the role of healthcare professionals is not clear. How can they come into contact with this target group and support them?

Authors: Thank you for the feedback. We added text that hopefully addresses this suggestion (see lines 123-125).

... based on the commitment and competencies to support families during the care of the person and or their family, at different levels of health care.

Reviewer 2

Lines 87-90: this paragraph is not clear. How do the targets differ from the subscales?

Authors: Thank you for the feedback. We added the meaning of each target and subscale (lines 160-162).

Reviewer 2

Lines 112-113: do the authors mean that the characteristics of truck drivers found in the studies in the literature are those listed below?

Authors: Thank you for the feedback. We revised the sentence for greater clarity (see lines 180-185).

The literature pointed to the lack of studies addressing family health from the perspective of family members of road transport workers, exploring the worker's perspective on their lifestyle, risk of diseases [20], work-life-family balance [9,21,22], psychological stressors, such as loneliness and loss of family conviviality [23], and the relationship with the family and coping strategies to overcome family isolation [24].

In this study, the characteristics of the families of road transport workers were described from studies carried out with LHTDs...

Reviewer 2

Lines 128-129: what does it mean that the study presents “previously unpublished factors”? Also, “are associated with or play a role in” would be more appropriate than “interfere”.

Authors: Thank you for the feedback. We edited the sentence for clarity (see lines 257-258).

This study presents factors that are associated with the perceived stress and family effectiveness of Brazilian family members of road transport workers.

Reviewer 2

Lines 130-131: this sentence seems incomplete.

Authors: Thank you for the feedback. We edited the sentence (see lines 259-260).

stressors that interfere with the health of road transport workers [2, 7-9, 21-23]. For this...

Reviewer 2

Lines 132 and 135: statements are made with two different objectives. Perhaps they should be summarised in two different hypotheses.

Authors: Thank you for the feedback. We added a second hypothesis (see lines 260-267).

For this study, the hypothesis are 1) The level of stress and family effectiveness of the family members of road transport workers will be influenced by social determinants of health such as years of schooling, income, diseases and number of people in the family; 2) The level of stress and family effectiveness of the family members of road transport workers will not be influenced by social determinants of health such as years of schooling, income, diseases and number of people in the family. This study aimed to evaluate the perceived stress level and the Family Effectiveness of family members of road transport workers.

Materials and Methods

Reviewer 2

Line 143: since the authors defined “family” as members who did not necessarily have to be related by blood, was this criterion applied when selecting the sample?

Authors: Thank you for the feedback. We added this information and edited this section for clarity (see lines 274-280).

The inclusion criteria in the study were to be a family member of road transport worker as a driver of an urban bus, an intercity and/or long-haul truck driver. Family members self-identified as family of LHTDs (and these family members could have been related by blood or could not have been related by blood). Family members had to be 18 years  of age or older and could read in Portuguese. The exclusion criteria were to be a family member of a retired road transport worker, orphan or widower, a family member 17 years or younger, or a non-identified family member of a LHTD.

Reviewer 2

Line 147: why was only one person per family selected?

Authors: Thank you for the feedback. We added this information (lines 287-288).

The participants were only one family member per family, which had an individual appointment for health care.

Reviewer 2

How many potential candidates refused to participate in the study?

Authors: Thank you for the feedback. We added this information (see lines 299).

Although precise numbers were not recorded, we estimate that approximately 10% of potential candidates declined to participate due to the time required to complete the questionnaire.

Reviewer 2

What was the minimum number of samples considered necessary to perform the analyses?

Authors: Thank you for the feedback. We added this information (See lines 355-357).

For the sample analysis, we estimated around 40 as the minimum number of participants, considering the number of family members in the health service each month.

Reviewer 2

Line 159: this section can be labelled “Measures" as the data collection actually starts in the paragraph above.

Authors: Thank you for the feedback. We added a subheading for Measures (see line 358).

2.2. Measures

Reviewer 2

For each of the scales used in the study, a sample item should be provided as well as the Cronbach’s alpha calculated in the study sample.

Authors: Thank you for the feedback. We added examples and Cronbach’s alpha for each scale (see lines 366, 384-385, 370-371, 393-395).

The Perceived Stress Scale (PSS) was developed and validated for use in Brazil [30], had its reliability obtained by internal consistency, in which Cronbach’s α=0.82

last month, how often have you been upset because of something that happened unexpectedly?

The instrument of ASF-E was developed in Detroit, MI for use with multicultural families at various educational levels, had its reliability obtained by internal consistency, in which Cronbach’s α ranged from 0.60 to 0.80.

As for example, the options in item 1 of the instrument and their respective options: There is anger or sadness in our family #1; People in our family do not openly express their feelings #2; Our family is happy, in general #3

Reviewer 2

Lines 190 and 196: these two sentences can be combined into one sentence.

Authors: Thank you for the feedback. The first sentence was deleted (see lines  393-395).

For example, the options in item 1 of the instrument and their respective options: “There is anger or sadness in our family” #1; “People in our family do not openly express their feelings” #2; “Our family is happy, in general” #3.

Reviewer 2

Line 104: this paragraph should be called “Data analysis”.

Authors: Thank you for the feedback. We added a subheading as requested (see line 454).

2.3 Data analysis

Reviewer 2

Line 218: there is a reference to an ethics committee, but it is not stated which institution.

Authors: Thank you for the feedback. We added this information (see lines 468-471).

The study was approved by the Ethics Committee of Federal University of Pelotas, Nursing Faculty (protocol code 5,892,602 and Certificate of Presentation for Ethics Assessment number 66722622.9.0000.5316, on February 14th, 2023.

Results

Reviewer 2

Line 230: it’s not clear what the information on Covid-19 vaccination doses does. Was this used in the analysis?

Authors: Thank you for the feedback. This sentence was deleted because it was not analyzed.

Reviewer 2

Using the full item descriptions in the tables would be more useful than the numbers.

Authors: Thank you for the feedback. We believe the tables are self -explanatory and facilitate the visualization and understanding of the results, so we would like to keep the figures in the text. If we misunderstood this reviewer’s request, please clarify and we would be happy to address it.

Discussion

Reviewer 2

Line 306: this first statement should be reformulated. The aim of the study was to investigate which factors – from a series of selected factors – are associated with the two main variables. Other factors that were not considered could also play an important role.

Authors: Thank you for the feedback. We edited this sentence for clarity (see lines 593-597).

This study presented how select factors -- years of schooling, family income, the number of members of the family group, the occurrence of diseases and the type of activity performed by the worker, as a driver -- influence perceived stress level and the family effectiveness of family members of road transport workers.

Reviewer 2

Line 316: the sentence is not clear. What is meant by “the study time”?

Authors: Thank you for the feedback. This sentence was edited from “study time” to “The years of schooling” (see line 605).

Reviewer 2

Lines 316-318: the two sentences are not well connected. If the higher level of education was found to be a predictor of stress, what possible reasons do the authors give for this result?

Authors: Thank you for the feedback. This sentence was modified (lines 607-624).

Considering that stress is a natural human biological response that prompts us to address challenges and threats in our lives, it is possible that overloading people's lifestyle activities with more years of schooling influenced their perception of stress level.

Reviewer 2

Lines 333-335: this sentence needs to be reformulated.

Authors: Thank you for the feedback. This sentence was revised (see lines 639-641).

The Control target aims to reduce anxiety, prevent and eliminate events that threaten family stability, and therefore break congruence or harmony [14].

Reviewer 2

Lines 363-366: is there data from studies that refer to the same target group or that share similar characteristics?

Authors: Thank you for the feedback. This sentence was edited for clarity (see lines 671-674).

A previous study reported higher levels of depression, anxiety, and stress, and whose family members were also more concerned about their partner’s well-being compared to community mothers [35].

Reviewer 2

Line 423: the sentence is not clear.

 Authors: Thank you for the feedback. This sentence was modified (see lines 671-674).

A previous study reported higher levels of depression, anxiety, and stress, and whose family members due to worker activity in remote places, family health, revealed high levels of depression, anxiety, were also more concerned about their partner’s well-being compared to community mothers [35].

Reviewer 2

Limitations

Limitations include the small sample size and the purely descriptive analysis. Further studies should take into account additional factors that would allow deeper statistical analysis. Another limitation is the cross-sectional design. Longitudinal studies would also be appropriate to gain a deeper understanding of the phenomenon. Diary studies, for example, could also be suitable for smaller samples.

Authors: Thank you for the feedback. This sentence was edited (see lines 845-846).

The limitations of this study are the small sample size, the descriptive analysis, the cross-sectional design. Another limitation...
